



# Identifying controls of extratropical cyclone intensity at genesis time and during intensification in the North Atlantic and Europe

Joona Cornér, Clément Bouvier, and Victoria A. Sinclair

Institute for Atmospheric and Earth System Research / Physics, Faculty of Science, University of Helsinki, Helsinki, Finland

**Correspondence:** Joona Cornér (joona.corner@helsinki.fi)

**Abstract.** Extratropical cyclones (ETCs) are an important part of the atmospheric circulation, cause most of the day-to-day weather variability, and have societal impacts through strong winds and heavy precipitation in the mid-latitudes. Therefore, from both weather forecasting and climate change perspectives it is crucial to understand how they develop and intensify. In this study we aim to identify which environmental background conditions, here called ETC precursors, have the most control

on the intensity of ETCs in the North Atlantic and Europe in the cold season. We apply an ensemble-based statistical method with ERA5 reanalysis data to associate climatologically typical perturbations in multiple ETC precursor fields at genesis time to distributions of five ETC intensity measures at time of maximum ETC intensity. We find that higher ETC wind intensity is associated with a stronger jet stream, especially downstream of the ETC centre, and increased meridional temperature gradients, with an emphasis on warmer upper levels south of the ETC centre. Precipitation is controlled by temperature and

moisture throughout the tropospheric column, with higher values associated with more precipitation. We perform the same analysis for four groups of ETCs with different average intensities and show that while differences exist in the controlling precursors among the groups, no clear patterns are observed. Due to the non-linear growth of ETC intensity, the precursor fields at genesis time offer limited explanations about differences in ETC intensity. Through analysing the temporal evolution of the four ETC intensity groups, we conclude that to understand differences in ETC intensity, as quantified by the five intensity

measures, it is necessary to investigate multiple ETC precursor fields and their time evolution.

## 1 Introduction

Extratropical cyclones (ETCs) are responsible for most of the variability in weather in the mid-latitudes and can have damaging impacts on infrastructure and society through strong winds and heavy precipitation (Gliksman et al., 2023; Hawcroft et al., 2012). They are also an important part of the atmospheric circulation of the climate system via transportation of heat, moisture,

and momentum polewards from low latitudes (Hartmann, 2015). Due to their importance to the climate system and impacts to society, ETCs and their development have been an active part of dynamic meteorology research for more than a century now (Bjerknes and Solberg, 1922). Understanding and being able to quantify the links between conditions in which ETCs develop and their intensity is essential for both weather forecasting purposes and the representation of ETCs and their potential impacts in future climate projections. Early research by e.g. Charney (1947) and Eady (1949) established that the main driver

of ETCs is waves forming due to baroclinic instability which rises from meridional temperature gradients on the polar front –



still an active area of research (e.g. Schemm and Rivière, 2019; Besson et al., 2021). Petterssen et al. (1955) also considered the importance of upper tropospheric vorticity advection on extratropical cyclogenesis. Later, in addition to this so-called dry-dynamic forcing, the contribution of diabatic effects on ETC development was understood, especially for rapidly intensifying cases (Wernli and Gray, 2024, and references therein).

The move from theory to diagnostic and prognostic tools arose with the invention of quasi-geostrophic theory and "potential vorticity (PV) thinking", which require assumptions that are valid in typical mid-latitude synoptic scale conditions (Charney, 1948; Hoskins et al., 1985; Holton and Hakim, 2013). Petterssen and Smebye (1971) and Deveson et al. (2002) used quasi-geostrophic diagnostics to identify different types of ETC life cycles based on the contribution of forcing from upper vs. lower levels. Their formulated ETC types have been shown to have differences in their intensification and preferred genesis and

occurrence areas (Gray and Dacre, 2006; Dacre and Gray, 2009). Using a large sample of ETCs, Plant et al. (2003) showed that on average the relative vorticity of an ETC depends on the ratio of upper-to-lower-level forcing, but that this relationship depends on meteorological conditions in which the ETC develop. Specifically, ETCs originating from frontal waves are more intense with larger lower-level forcing while low pressure systems not clearly associated with frontal waves are more intense when upper-level forcing dominates.

Following the prevalence of PV thinking a technique called piecewise PV inversion, which attributes contributions of PV anomalies to the wind field in different levels of the atmosphere, was developed (Hoskins et al., 1985; Davis and Emanuel, 1991). For example, Seiler (2019) used piecewise PV inversion to show that in a majority of ETCs in the Northern Hemisphere lower-level PV generated via condensational heating contributes the most to intensification as measured by the $850\,\mathrm{hPa}$ relative vorticity. Dolores-Tesillos et al. (2022) showed that the latent heat release, which is projected to increase in the future (Catto

et al., 2019), is likely to increase the size of an ETC's wind footprint. Piecewise PV inversion was also used by Plant et al. (2003) along with Ahmadi-Givi et al. (2004) to show that the effects of diabatic heating need to considered in the ETC life cycle types and that latent heat release can e.g. weaken upper-level PV anomalies, thus affecting coupled lower-level features as well (Madonna et al., 2014).

While these kind of analyses offer detailed insight into the dynamics of ETCs, they require certain assumptions or balance

conditions, are limited to specific variables, and can be computationally relatively expensive to perform for large samples of ETCs. A next step to relate ETC development dynamics to their intensity with a more climatological perspective came from weather forecasting applications. These methods are able to utilize the more direct attribution of anomalies to changes in intensity, akin to piecewise PV inversion, while allowing a large sample to be analysed without very expensive computations. These kind of methods are categorized as sensitivity analyses and they work by estimating how an initial perturbation affects a

forecast field throughout a weather forecast (e.g. Martin and Xue, 2006). Hakim and Torn (2008) and Torn and Hakim (2008) applied sensitivity analysis to an ensemble forecast and used sample statistics to calculate the sensitivity of forecast fields to initial state perturbations. They called this method ensemble-based sensitivity analysis or ensemble sensitivity analysis (ESA). ESA is formulated in a way which allows it to be used for identifying climatological sensitivity relationships, i.e. quantifying changes associated with typical perturbations in the initial fields. Ancell and Hakim (2007) showed that ESA gives similar

sensitivity results as adjoint sensitivity analysis, which tracks the initial perturbation through time with a tangent-linear model,





for the mean sea level pressure of an ETC 24 h after the initialization of the forecast. Ensemble sensitivity had, however, more emphasis on synoptic scale features in its sensitivity fields compared to adjoint sensitivity.

In addition to its lack of required assumptions, the benefit of ESA arises from the use of sample statistics which allows one to investigate the sensitivity of any forecast field to any given initial state variable, as long as there is enough homogeneity
in the initial state among the ensemble members. For example, ESA can be used to study the influence of upper-level fields, lower-level fields, or proxy fields for latent heat release such as moisture on ETC intensity. Garcies and Homar (2009) utilized the customizability of ESA and furthered its application by using atmospheric fields from reanalysis data instead of ensemble forecasts. In their framework, the ensemble consisted of Mediterranean cyclones categorized as similar according to a $k$-means cluster analysis. The perturbations in the ensemble were thus represented as deviations from a climatological mean of each
cluster. They found that minimum sea level pressure in Mediterranean cyclones is sensitive to atmospheric structures over Western Europe, northern Africa, and parts of east North Atlantic 24 h before. Like Ancell and Hakim (2007), Garcies and Homar (2009) compared ESA to adjoint sensitivities and found similarities between the two, with the former emphasising synoptic-scale features. Dacre and Gray (2013) used ESA with reanalysis data to investigate how precursors conditions of ETCs affect their intensity in the North Atlantic. They had fewer categories than Garcies and Homar (2009) – location-based
west and east North Atlantic ETCs – but they introduced homogeneity in the ensembles through a Lagrangian approach by centering the ETCs around a common reference point, namely the centre of the ETC. They found that 850 hPa relative vorticity is sensitive to upper-level troughs upstream of the ETC centre in both west and east North Atlantic ETCs but only west North Atlantic ETCs were sensitive to low-level thermal gradients. East North Atlantic ETCs were, however, sensitive to diabatically generated PV anomalies in the middle troposphere. A similar, more recent approach studying ETCs which affect northern
Europe was used by Laurila et al. (2021) who split their set of ETCs into windstorms and non-windstorms based on a threshold 90th percentile of 10 m wind gust and also used cyclone-centered composites to ensure homogeneity in the ETC precursor fields. Firstly, they showed that mean sea level pressure and maximum 10 m wind gust of windstorms in northern Europe are more sensitive to precursors during the cold season than the warm season. Partly contradictory to the results of Dacre and Gray (2013) for west North Atlantic ETCs, Laurila et al. (2021) also showed that the intensity of windstorms in northern Europe
is associated the strongest with the magnitude of the lower-level frontal gradient and less so with moisture and upper-level features. This discrepancy could, however, be attributed to the more restricted geographical domain, split into cold and warm season ETCs, and the investigation of only windstorms in Laurila et al. (2021). Dacre et al. (2019) used ESA to investigate precipitation and integrated vapour transport associated with intense ETCs and found that they are the most sensitive to total column water vapour on the south side and downstream of the ETC centre 24 h before – a result shared by Laurila et al. (2021)
for mean sea level pressure as the metric for intensity.

The aim of this study is to identify which meteorological factors at genesis time control the maximum intensity of ETCs the most in the North Atlantic and Europe during the extended cold season. To achieve this, the ESA method is used in a way adapted from previous research. The novelty comes from the following approaches: 1. ESA is applied between multiple ETC precursors and multiple intensity measures. The investigated intensity measures were identified to represent the variability of
ETC intensity comprehensively by Cornér et al. (2025). 2. In addition to measures which describe the dynamical intensity of





an ETC, the analysis includes intensity measures which have been shown to quantify storm-caused damages reasonably well (Karremann et al., 2014; Roberts et al., 2014; Moemken et al., 2024). 3. Sensitivities are calculated separately for groups of ETCs which differ in their intensity, as objectively identified by Cornér et al. (2025). The ultimate aim is to understand and explain the differences in the controls between ETCs of varying intensities based on their average structure.

This paper is structured in the following way. Section 2 presents the data and methods used in the study. Section 3 includes the results of the ensemble sensitivity analysis. Differences between ETCs of varying intensities are investigated in Section 4. Finally, Section 5 concludes the results.

## 2   Data and methods

### 2.1   ERA5 reanalysis

ERA5 reanalysis (Hersbach et al., 2020) is the most recent global reanalysis product produced by the European Centre for Medium-Range Weather Forecasts (ECMWF). This study uses ERA5 fields on its native reduced Gaussian grid with resolution N320 (or spectral truncation TL639) interpolated to regular intervals along parallels, which corresponds to a grid size of $0.281°/31\,\mathrm{km}$ at the equator. For data availability reasons a couple of variables are on a regular $0.25° \times 0.25°$ latitude–longitude grid. All fields in this study are sampled at a 3-hourly temporal resolution. The ERA5 data considered in this study cover

the extended boreal winter season (October–March) between the years 1979–2022. ERA5 is used for ETC tracks, intensity measures, and precursors.

### 2.2   ETC tracking, intensity measures, and precursors

#### 2.2.1   ETC tracking

ETCs are first tracked in the Northern Hemisphere with $850\,\mathrm{hPa}$ relative vorticity (VO) truncated to T42 resolution as input

to the TRACK feature tracking software (Hodges, 1994, 1995, 1999). The resulting ETC tracks are then limited to an area covering the North Atlantic ocean and parts of Europe; specifically, the ETCs need to have maximum VO along their track inside a box bounded by $80°\,\mathrm{W}$ to $40°\,\mathrm{E}$ in longitude and $30°\,\mathrm{N}$ to $75°\,\mathrm{N}$ in latitude. To exclude weak, short-lived, and stationary systems, further criteria include a minimum T42 VO value of $1 \times 10^{-5}\,\mathrm{s}^{-1}$, a minimum 2-day lifetime, and a minimum travelled distance of $1000\,\mathrm{km}$. Additionally, to include only intensifying ETCs, the maximum VO value cannot

occur earlier than $24\,\mathrm{h}$ after ETC genesis. The total number of ETC tracks identified and meeting the criteria in the 43 extended winter seasons is 7361. Further details of the tracking can be found in Cornér et al. (2025).

#### 2.2.2   ETC intensity measures

Cornér et al. (2025) analysed 11 ETC intensity measures associated with the ETC tracks. The ETC intensity measures were analysed as point-per-track values, i.e. each ETC track had one value per intensity measure associated with it. The measures

included dynamical measures and what they called "impact-relevant" measures. They performed analyses to identify measures





which describe ETC intensity comprehensively and non-redundantly, and found that five intensity measures, namely $850\,\mathrm{hPa}$ relative vorticity (VO), $850\,\mathrm{hPa}$ wind speed (WS850), a wind footprint (WFP), a storm severity index (SSI), and precipitation (PRECIP) do that. The main focus in this study is on these five measures for which the detailed calculation is described in Cornér et al. (2025). However, other intensity measures used in Cornér et al. (2025) are included in parts of the analysis in this study as well.

### 2.2.3 ETC precursors

ETC precursors are selected based on theories concerning the development and growth of baroclinic waves and ETCs. Investigated precursors include wind speed at $300\,\mathrm{hPa}$ (WS300), potential vorticity anomaly at $300\,\mathrm{hPa}$ (PVa300, anomaly from $30\,\mathrm{d}$ running mean values as in Dolores-Tesillos et al. (2022)), temperature at 300 and $850\,\mathrm{hPa}$ (T300, T850), temperature anomalies (from cyclone-centred composites) at 300 and $850\,\mathrm{hPa}$ (Ta300, Ta850), $2\,\mathrm{m}$ temperature (2T), total column water vapour (TCWV), and lower tropospheric lapse rate (GAMMA) which here is defined as

$$\Gamma = -\frac{T(850\,\mathrm{hPa}) - T(500\,\mathrm{hPa})}{Z(850\,\mathrm{hPa}) - Z(500\,\mathrm{hPa})}, \tag{1}$$

where $T$ is temperature and $Z$ is geopotential height. Unlike the ETC intensity measures which have one value per track, the precursors are analysed on a spatially two-dimensional grid around the ETC centre point (the VO maximum). The creation of the precursor grid is described in Sect. 2.3.

### 2.3 Cyclone composites

Cyclone compositing is a method to investigate the mean structure and temporal evolution of a sample of cyclones by averaging meteorological fields in the area surrounding the cyclones (Catto et al., 2010). By averaging (or compositing) a large sample of cyclones it is possible to study the main features or processes occurring in cyclones with a manageable amount of information as averaging eliminates noisy and non-robust features. As long as the cyclones composited together are similar enough, the resulting composite retains relevant information about the sample of cyclones. However, the composite does not necessarily represent a single instance of cyclones included in the composited sample. There are various steps which can be taken to ensure physical and dynamical consistency in the sample of cyclones before compositing. The ones used in this study are detailed below.

The cyclone compositing used in this study is adapted from the method described in detail in the appendix of Bengtsson et al. (2007) and used in e.g. Catto et al. (2010), Dacre et al. (2012), Sinclair et al. (2020), and Dolores-Tesillos et al. (2022). The method produces cyclone-centered composites in which the frame of reference is the ETC centre point (here defined as the tracked T42 VO maximum) at a specific point in time relative to the ETC life cycle (e.g. time of genesis or time of maximum VO). Furthermore, before the averaging, the selected fields are rotated to make the direction of travel the same for all ETCs. This normalises the location of cyclone-relative features such as fronts and air streams.

Specifically, the compositing is performed as follows. First, a radial coordinate system with a given radius is centered at the pole. In this study the radius is set at $18°$. Second, for each ETC this radial grid is relocated to be centered at the point of





interest – the ETC vorticity centre at a selected time during the ETC track. Third, the data on the original grid are interpolated onto the radial grid. In this study the radial grid resolution is $0.5° \times 0.25°$ which corresponds to 720 and 72 grid points in the azimuthal and radial directions, respectively. In the zonal and meridional directions the radial resolution is close to the resolution of ERA5 ($0.281°$), which should minimize the effects of interpolation. Fourth and finally, the interpolated data on the radial grid are rotated to make the direction of propagation consistent between ETCs in the sample. Here, after the rotation, direction of propagation is eastward which corresponds to the azimuthal direction of $0°$ on the radial grid, as in the unit circle. The resulting interpolated and rotated fields are averaged together to create the composites. Unless otherwise mentioned, all further references to compass directions in composited fields are relative to this rotated frame of reference in which east is the direction of propagation.

As opposed to projections on a map, the use of a rotated radial grid can reduce distortion in the composites (Wang and Rogers, 2001; Catto et al., 2010). Depending on the interpolation, it can also make comparisons between composites from different datasets easier. This is because the direction of propagation, which can vary for tracks of the same cyclone from different datasets, is normalised in the process.

## 2.4  Ensemble sensitivity analysis

In this study, we combine the cyclone-centered approach of Dacre and Gray (2013) and Laurila et al. (2021) with using statistically derived groups of cyclones as in Garcies and Homar (2009). As in Dacre and Gray (2013), Dacre et al. (2019), and Laurila et al. (2021), the ESA utilizes the fields used to create the cyclone composites. In fact, the ensemble consists of the sample of ETCs which is used to create the composites. For each point $ij$ in a two-dimensional field of an ETC precursor $x$, the sensitivity is calculated against a spatially independent response function $J$ which in this study is any of the intensity measures. Each ETC is a member in an ensemble of size $M$ which equals the size of an ETC intensity group or the number of all ETCs for the whole track dataset. The first ingredient and the basis of the sensitivity is the linear regression coefficient or slope which is defined as

$$\beta_{ij} = \frac{\partial J}{\partial x_{ij}}. \tag{2}$$

The linear regression coefficient $\beta$ is also called raw sensitivity.

The raw sensitivity values are multiplied by the standard deviation $\sigma_{ij}$ of the precursor field. This counteracts the effect of geographical variability of the variance of $x_{ij}$ in Eq. 2 to yield a climatological sensitivity value (Garcies and Homar, 2009). Additionally, standardizing the raw sensitivities with the standard deviation makes the units of the sensitivity the same as the response function, increasing interpretability and comparability between different precursor fields. The sensitivity is thus given by

$$S_{ij} = \beta_{ij}\sigma_{ij}. \tag{3}$$

Finally, as in Dacre et al. (2019), statistically insignificant signals are filtered out by using the false discovery or detection rate (FDR) method by Wilks (2016) which corrects the significance level of multiple statistical hypothesis tests performed




simultaneously – in this case in multiple locations on the cyclone-centered grid. According to this method the corrected $p$ value considered to be statistically significant is the largest $p$ value which satisfies the condition $p_n \leq n\alpha/N$, where $p_n$ is the $n$th smallest $p$ value of $N$ hypothesis tests and $\alpha$ is the control significance level (i.e. the significance level for a single hypothesis test ($N = 1$)). Like in Dacre et al. (2019), here the uncorrected statistical significance level is set to $\alpha = 0.1$.

The significant sensitivity values indicate how large the change in the response function associated with a climatologically
typical perturbation is, i.e. a perturbation of one standard deviation in the precursor field. Specifically, the interpretation of the sensitivity values is with respect to an increase of one standard deviation in the precursor field. However, the interpretation of negative sensitivity values can be inverted. A decrease of one standard deviation in the precursor field is associated with an increase in the response function whose magnitude is given by the absolute value of the sensitivity. Since the precursors are analysed on a cyclone-relative grid, their "climatologies" are determined by the mean and variability of the composites and
are thus Lagrangian in nature. It should also be noted that sensitivity implies only correlation, not causation, and any physical mechanisms behind a sensitive relationship need to be deduced afterwards.

In this study the sensitivities are calculated between precursors at time of genesis and intensity measures at the time of maximum VO (or for PRECIP, $12\,\mathrm{h}$ before). Analysing the sensitivity of intensity to environmental conditions at genesis ensures that the effect of the ETC's own circulation is minimized. Furthermore, it offers the best potential for understanding
controls on ETC intensity from a forecasting perspective. Unlike in previous studies in which sensitivities were calculated for example $48\,\mathrm{h}$ before time of maximum intensity (e.g. Laurila et al., 2021) or $48\,\mathrm{h}$ after genesis (e.g. Dacre and Gray, 2013), the time difference between genesis and maximum intensity is naturally variable between ETCs. It is, however, a statistically consistent time period.

### 2.5  ETC intensity grouping

In this study ETCs are split into groups based on multiple intensity measures. Specifically, the grouping is based on the sparse principal component analysis (sPCA) which was used in Cornér et al. (2025) to identify the five intensity measures mentioned in Sect. 2.2.2. Sparse PCA is a form of principal component analysis in which the principal components (PCs) do not represent the explained variance in the dataset as a linear combination of all the input features. Instead, it forces the weights of features with less explainative power to zero to improve interpretability of the PCs. To obtain the intensity groups, we first split the
sPCA feature space into four quadrants which contain non-average values of the first two principal components shown in Fig. 1. Specifically, we remove ETCs whose two-dimensional coordinate values (PC1;PC2, with a precision of $0.001$) fall inside an area in the sPCA space defined by

$$y = \pm \frac{1}{\alpha|x|}, \tag{4}$$

where $x$ is the PC1 value, $y$ is the PC2 value, and $\alpha$ is a scaling parameter. The four groups then consist of the remaining
ETCs which occur in quadrants defined by different combinations of positive and negative values of PC1 and PC2. The value of $\alpha$ was determined based on the proportion of tracks retained and the average minimum Euclidean distance between points in adjacent groups after the application of Eq. 4. The chosen value is $\alpha = 50$, which retains 3059 ($42\,\%$) of all tracks. Weights





of the 11 intensity measures in the PCs are shown in Fig. S1. The weight of PC1 is composed mainly of wind speed measures (wind speed at $850\,\mathrm{hPa}$, $925\,\mathrm{hPa}$, $10\,\mathrm{m}$, and wind gust at $10\,\mathrm{m}$), and thus the axis is labelled qualitatively from "Calm" (PC1
$< 0$) to "Windy" (PC1 $> 0$). Similarly, the weight of PC2 is composed mainly of precipitation measures (an instantaneous and an accumulated precipitation value), and thus the axis is labelled qualitatively from "Dry" (PC2 $< 0$) to "Rainy" (PC2 $> 0$). This produces four ETC intensity groups named "Rainy+Calm", "Rainy+Windy", "Dry+Calm", and "Dry+Windy".

The grouping process as well as its result is shown graphically in Fig. 1. Due to the unsymmetrical shape of the sPCA projection, the number of ETCs in each group varies. The most tracks are in the two groups in which the signs of PC1 and
PC2 values are the same, i.e. Dry+Calm and Rainy+Windy. It should be noted that while the application of Eq. 4 removes a majority of the "most average" ETCs from the sample, the qualitative labels given to the groups do not indicate that all ETCs in the groups have extreme values of both windiness and precipitation. However, the shape of the filtering function ensures that an ETC has "extreme" values in either PC1 or PC2, or more moderate, closer-to-average values in both PC1 and PC2 at the least.

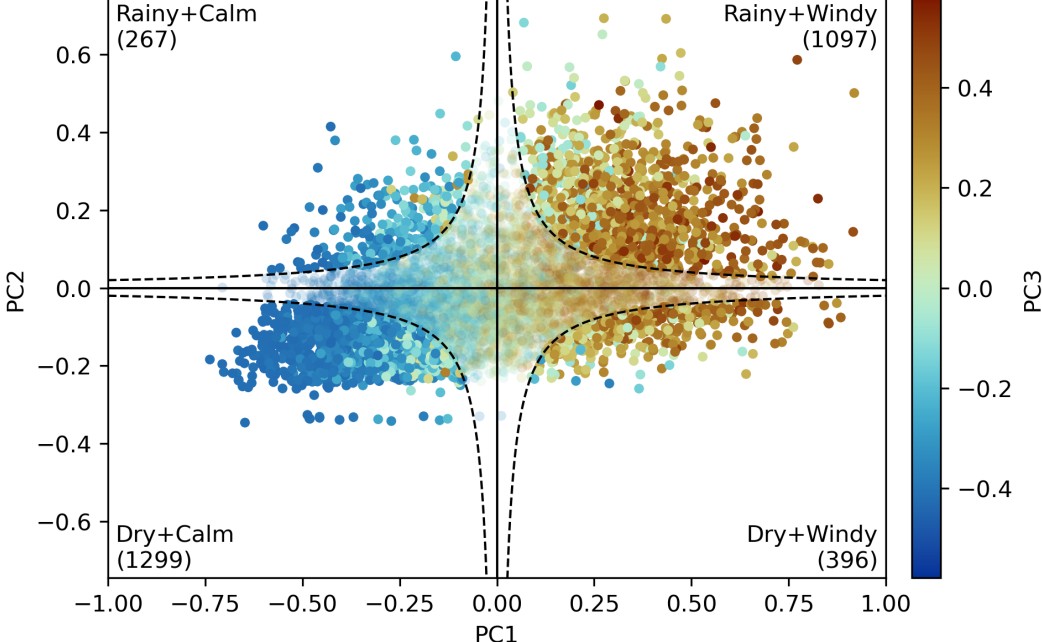

**Figure 1.** ETCs projected in the sPCA feature space. The dashed lines show curves defined by Eq. 4. Points in the blurred area are "close-to-average" ETCs and thus omitted from the intensity groups. The names and sizes of the intensity groups are written in the quadrants. The colour of the points indicates the PC3 value (mainly WFP) which is not used in the grouping. The weights of the input features in the principal components are shown in Fig. S1. For reference, the ETC with the smallest Euclidean distance from origin of the sPCA space and thus the most "average" ETC has a WS850 value of $30.2\,\mathrm{m\,s^{-1}}$ and a PRECIP value of $2.3\,\mathrm{mm\,(3\,h)^{-1}}$.





When calculating sensitivity values for the four intensity groups, the standard deviation used in Eq. 3 is the standard deviation of the sample of the groups together which amounts to $42\%$ of all ETCs. Since the standard deviation values used are the same for each subsample, i.e. an intensity group, the sign and relative strength of sensitivity in a subsample is determined by the regression coefficient. This choice enables fair comparison of sensitivity values between the groups. Moreover, the removal of average-intensity ETCs from the full dataset does not greatly affect the composite standard deviation patterns of the precursors

in the reduced sample. They remain very similar to the full dataset and only the magnitude changes slightly (not shown).

     Figure 2 shows the ETC track density distributions of the four intensity groups. A clear divide is seen between the Windy and Calm groups in track density. The Windy ETCs occur mostly over the ocean along the main North Atlantic storm track whereas the Calm ETCs occur mostly either over land, the Mediterranean sea, or the edges of the storm track. Between the Windy groups, Rainy+Windy ETCs occur mostly at the start of the storm track close to the eastern coast of North America,

more in the southern parts of the domain (Fig. 2b). Dry+Windy ETCs occur more towards the end of the storm track close to Iceland (Fig. 2d) which is northeast of the Rainy+Windy maximum due to the southwest–northeast tilt of the storm track. A similar north–south divide is seen between groups Rainy+Calm and Dry+Calm. Group Rainy+Calm has maxima in track density in the southern parts of the North Atlantic and western Mediterranean (Fig. 2a). The area of large track density over the ocean is oriented zonally as opposed to the northeastern tilt seen in the Windy groups. This indicates that these ETCs do not

necessarily travel along the main storm track. Track density in group Dry+Calm has four local maxima in western and eastern Mediterranean, northeastern Europe, and the Hudson Bay area (Fig. 2c). These results show that in the Mediterranean most Rainy ETCs occur in the western parts and that in northern Europe there are few Rainy ETCs.

## 3 Sensitivity to ETC precursors

### 3.1 Sensitivities for all ETCs

The sensitivity signals between the five relevant intensity measures and all investigated ETC precursors for an ensemble of all 7361 ETCs are shown in Fig. 3. In qualitative terms the sensitivity signals for all four wind-based intensity measures (VO, WS850, WFP, and SSI) are very similar. The most striking features associated with more intense winds in ETCs are stronger WS300 downstream, higher T300 on the south side, and lower T850 and 2T on the north side of the ETC centre. The signals from GAMMA reflect the changes in temperature at both upper and lower levels, with smaller values of GAMMA

overall (upper troposphere warming and lower troposphere cooling, decreasing the lapse rate – a more stable mid-troposphere) associated with more intense winds. The sensitivity pattern of TCWV mimics that of the lower-level and surface temperature fields with an increased gradient associated with stronger winds. Given the composite structure of WS300 at genesis time (grey contours in Fig. 3) with the highest wind speeds vertically aligned with the ETC centre, the larger WS300 sensitivity values downstream of the ETC centre indicate that the wind intensity will be stronger if the upper-level jet is stronger but also if

the ETC is initially located at the right entrance of a jet streak. Both of these factors align with previous knowledge of ETC intensification based on cyclogenesis due to vorticity advection according to quasi-geostrophic theory (Holton and Hakim, 2013). Through thermal wind balance, the positive sensitivity signal from WS300 is also indicative of increased baroclinicity



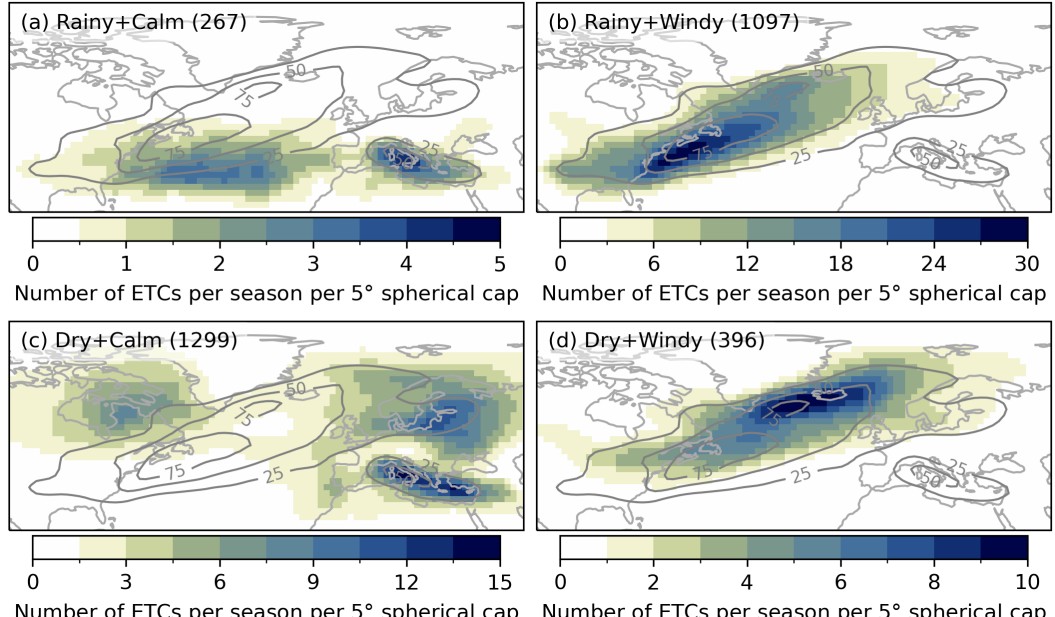

**Figure 2.** Track densities of the four intensity groups. The value indicates the number of ETCs occurring within a spherical cap with a radius of $5°$ geodesic per season (six months). The contours show the track density distribution of the full dataset (7361 ETCs) in the same unit. Total number of ETC tracks in each group is shown in the parentheses. Note the different values in the colourbars.

being associated with stronger ETCs. This is in line with the signals from all three temperature precursors showing an increased meridional temperature gradient being associated with more intense ETCs.

The sensitivity signal for PRECIP differs from the wind-based intensity measures. Sensitivity of PRECIP is clearly dominated by the three temperature fields and TCWV. Larger PRECIP values are associated with overall warming of the troposphere with only positive sensitivity values appearing in the temperature fields. TCWV follows the same positive pattern, which is to be expected based on the Clausius–Clapeyron relation. Moreover, precipitation in ETCs occurring at high latitudes is limited by moisture availability, among other factors (Pfahl and Sprenger, 2016). Therefore, moist air masses (relative to climatology) can

be expected to lead to more precipitation especially at high latitudes. Noteworthy is that the sensitivity signal is the strongest southeast of the ETC centre, implying a stronger thermal ridge, which could indicate links to more moisture being transported in the feeder airstream to the warm conveyor belt (Dacre et al., 2019; Eckhardt et al., 2004), leading to more precipitation (Field and Wood, 2007). This sensitivity pattern is consistent with the one found by Dacre et al. (2019) and indirectly by Laurila et al. (2021) for TCWV. PRECIP is also weakly sensitive to GAMMA close to and on the south side of the ETC centre. These

sensitivity values are positive, which indicates more precipitation associated with an initially more unstable atmosphere.

    To aid comparison of the magnitudes of the sensitivity values between different precursors, we also introduce a method to compress the two-dimensional sensitivity field into a single number. This is done by calculating the mean value of the absolute





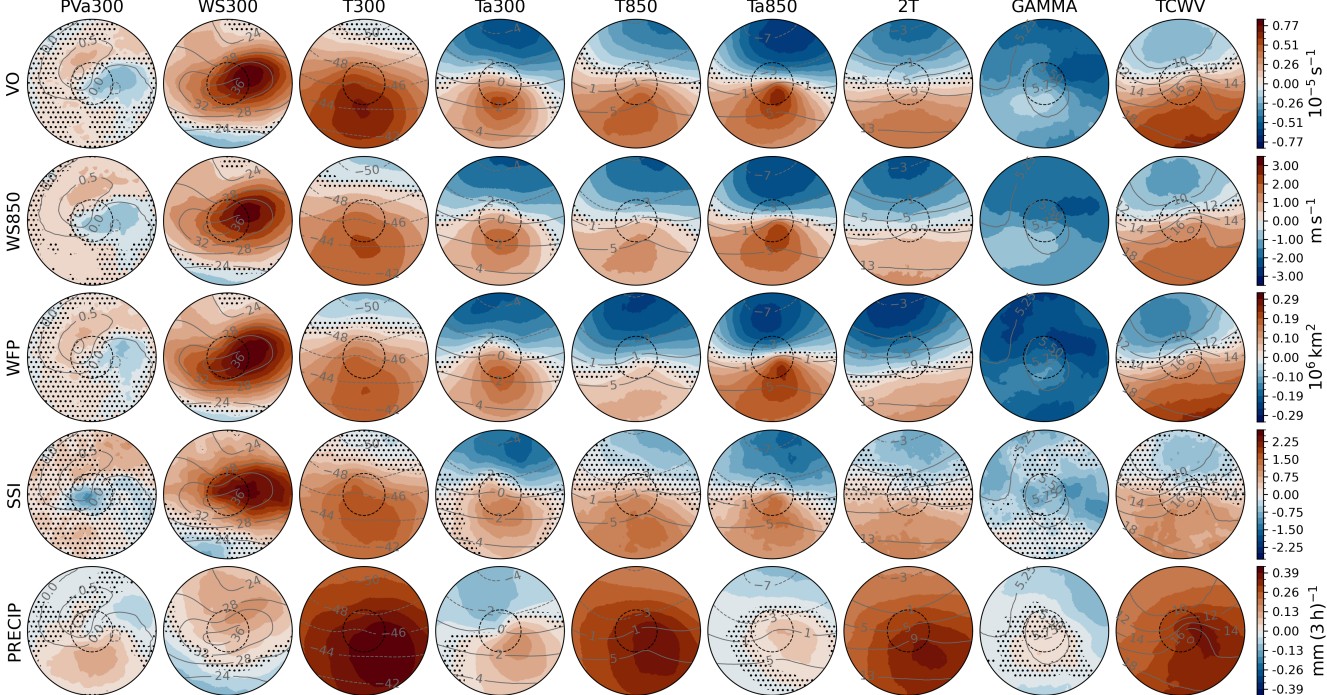

**Figure 3.** Sensitivity signal of each combination for all ETCs (shading) and the composite field of precursors at genesis (contours). Intensity measures stay constant along rows and precursors stay constant along columns. There is one colourbar per intensity measure, i.e. per row. The hatched areas indicate statistically nonsignificant values and the dashed black circle denotes a radius of $6°$ geodesic. The radius of the whole circle is $18°$ geodesic. Units of precursors are PVU ($10^{-6}\,\mathrm{K\,m^2\,kg^{-1}\,s^{-1}}$) for PVa300, $\mathrm{m\,s^{-1}}$ for WS300, $°\mathrm{C}$ for T300, Ta300, T850, Ta850, and 2T, $°\mathrm{C\,km^{-1}}$ for GAMMA, and $\mathrm{kg\,m^{-2}}$ for TCWV.

values of statistically significant sensitivity values. Statistical significance of the sensitivity values is determined with the FDR method as described in Sect. 2.4. The mean sensitivity value is weighted by the proportion of significant sensitivity values of all

points $ij$ in the field. If all values in the sensitivity fields are nonsignificant, the weight equals zero. Before taking the absolute values, information about the ratio of positive to negative significant sensitivity values is saved.

    By investigating these "collapsed" average sensitivity values shown in Fig. 4, we see the relative contribution of each precursor to each intensity measure in a concise manner. Based on this metric WS850 and SSI are the most sensitive to WS300, with WFP and VO having only slightly smaller sensitivity values. VO is the most sensitive to T300 and WFP to GAMMA.

The temperature precursors at $850\,\mathrm{hPa}$ and $300\,\mathrm{hPa}$ have different dominant effects for wind-based measures, i.e. wind intensity is more sensitive to T300 than Ta300 but more sensitive to Ta850 than T850. Based on the overwhelmingly positive T300 sensitivity signals for all intensity measures this could indicate that ETCs which have their genesis towards the start of North Atlantic storm track – which is climatologically warmer – develop stronger winds. It could also be linked to the genesis position relative to the jet, which is upstream of the jet maximum for intense ETCs, as discussed above. The larger sensitivity



to the strength of the frontal gradient at lower levels quantified by the temperature anomaly (Ta850) suggests that ETCs with
stronger baroclinicity at these levels or ETCs possibly developing from a frontal wave have stronger winds. Plant et al. (2003)
found the latter to be true for the synoptic–dynamic ETC classification types A and B of Petterssen and Smebye (1971) but not
for more diabatically driven type C ETCs of Deveson et al. (2002).

VO, WS850, and WFP alike are sensitive to all-around negative perturbations in GAMMA, i.e. a more stable mid-troposphere
at genesis is associated with stronger winds. WFP is especially sensitive to a decrease in GAMMA and thus an increase in sta-
bility. The reason for this may be related to the Rossby radius of deformation. The Rossby radius of deformation is directly
proportional to the stability of the atmosphere (Gill, 1982). Thus, smaller values of GAMMA, which indicate a more stable
atmosphere, would be associated with larger Rossby radii of deformation and allow for larger ETCs with larger areas with
strong winds. The negative sensitivity of VO and WS850 to GAMMA are, in turn, not consistent with a physical explanation
from theory. The Eady growth rate, a measure of baroclinicity, is inversely proportional to stability (Lindzen and Farrell, 1980;
Hoskins and Valdes, 1990). Therefore, stronger growth would be expected in conditions in which the atmospheric stability is
smaller (i.e. larger GAMMA), the opposite of what we see. Also, one would expect larger VO values in a less stable atmo-
sphere given a similar forcing for ascent based on the quasi-geostrophic omega equation (Holton and Hakim, 2013). According
to Besson et al. (2021), the effect from a small Eady growth rate is, however, larger than from omega forcing since large ETC
deepening rates are observed with small omega forcing and large Eady growth rate, but not vice versa.

It is also possible that the negative sensitivities are affected by not only directly ETC-related processes but by correlations
between the precursors. The opposing perturbations in temperature at different levels, i.e. a warm perturbation at upper levels
and a cold perturbation at lower levels would result in smaller values of GAMMA. This explanation is supported by the fact
that control from GAMMA is the strongest for WFP (Fig. 4) for which the negative sensitivity from T850 is the most prominent
out of VO, WS850, and WFP (Fig. 3).

For PRECIP there are not as many nuances in the sensitivity signals as for the wind-based measures, which can be seen in
both the full sensitivity fields (Fig. 3) and the "collapsed" sensitivity values (Fig. 4). PRECIP is the most sensitive to T300,
T850, TCWV, and 2T, respectively, with only positive values present in these fields. A precursor which does not have large
sensitivity values for either PRECIP or the wind-based intensity measures is PVa300.

To further understand and interpret the sensitivity signals for all ETCs, we can consider where geographically ETC genesis
in the four intensity groups occurs. Figure 2 shows were ETCs with certain intensity characteristics (e.g. Windy or Rainy)
preferentially occur and Fig. S2 from where they originate. This information can be combined with climatological distributions
of the precursors to understand their potential influence on the sensitivity fields. Here, climatology refers to mean values of the
extended winter season (ONDJFM) between the years 1979 and 2022. Climatological distributions of precursors discussed in
the following text are shown in the supplement (Fig. S3–S7).

For example, from the track density distributions in Fig. 2 it is evident that Windy ETCs occur mostly along the main storm
track. This is well inline with the results shown in Fig. 3 and 4 which indicate that large wind intensity is associated with
a strong jet stream. As discussed above, the sensitivity pattern from WS300 to wind intensity can be interpreted as ETCs
having their genesis more in the right hand entrance of a jet streak, with the largest sensitivity values downstream of the ETC





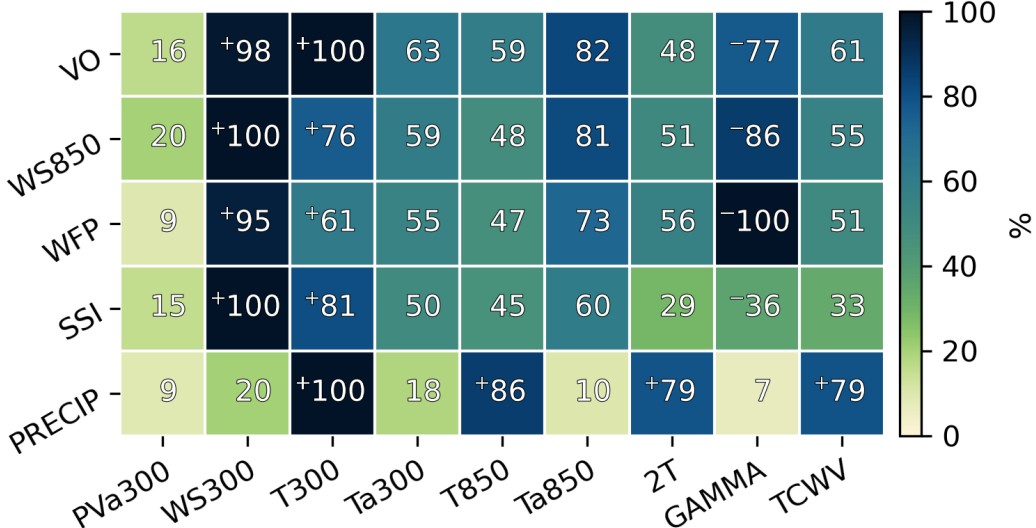

**Figure 4.** Proportions of average magnitudes of significant sensitivity signals of the maximum value (per intensity measure). Maximum average magnitudes are VO: $0.45 \times 10^{-5}\,\mathrm{s}^{-1}$, WS850: $1.68\,\mathrm{m\,s}^{-1}$, WFP: $0.19 \times 10^6\,\mathrm{km}^2$, SSI: 1.38, and PRECIP: $0.36\,\mathrm{mm\,(3\,h)}^{-1}$. The plus (minus) sign in front of a number indicates that at least 90 % of the significant sensitivity values are positive (negative).

centre. However, it is also possible to understand the pattern from a climatological perspective. ETCs which originate from the beginning of the storm track, i.e. mostly Windy ETCs, develop stronger winds (Fig. S2 and S3). The strong positive sensitivity signals from T300 support this, as the start of the storm track is climatologically warmer than the end due to the southwest–northeast tilt (Fig. S4). At lower levels in which the climatological temperature distribution tilts more towards northeast with the storm track (Fig. S5), the overall temperature distribution is less important, and it is the strength of the thermal gradient

(Ta850) – a more transient feature – which has more control on the wind intensity.

For PRECIP the correspondence of the ESA results with the track and genesis density distributions and the precursor climatologies is straightforward. Compared to their Dry counterparts, Rainy ETCs occur more in the southern parts of the domain which climatologically have more moisture available (Fig. S6) and higher temperatures. This is exactly what we see in the ESA as well (Fig. 4).

It should be noted that while climatological conditions likely affect the genesis and development of ETCs to some extent and ETCs contribute to the climatological mean state, for transient phenomena like ETCs, the genesis states deviate from the climatology from case to case by definition. This is evident in our results since the variation in a precursor field is as large as differences between the precursor composite means of the intensity groups (not shown). Moreover, the track and genesis density distributions (Fig. 2 and S2) have overlap between the groups and therefore do not represent unique climatological

conditions in the first place.





An example of this – a precursor genesis field not matching the climatological state – is GAMMA in the Windy intensity groups. Their genesis composites show relatively small GAMMA values (Fig. S8), which agrees with the results of the ESA (Fig. 4). However, the Windy ETCs have genesis hot spots in the Rocky Mountains (Rainy+Windy) and between Greenland and Iceland (Dry+Windy, Fig. S2), areas with climatologically large GAMMA values (Fig. S7). In fact, the area with the

smallest climatological GAMMA values in the domain, the Labrador Peninsula (Fig. S7 and Mbengue et al., 2019), has almost no occurrences of cyclogenesis (Fig. S2). Therefore, especially since the negative GAMMA sensitivities do not match theoretical expectations (as discussed above), their interpretation is autocorrelation with opposing perturbations in the temperature precursors at $300\,\mathrm{hPa}$ and $850\,\mathrm{hPa}$.

### 3.2 Sensitivities for ETC intensity groups

In the previous section (Sect. 3.1) we showed what the sensitivity patterns and the relative magnitudes of average sensitivities are in the full sample of tracked ETCs. In this section the results are shown for the ESA done separately for the four intensity groups derived in Sect. 2.5. The purpose is to investigate if there are differences in the sensitivities for ETCs of varying intensities and thus if the genesis conditions can be used to determine what type of ETC may develop.

Results of the ESA for the groups for each intensity measure are shown in the "collapsed" form in Fig. 5 and as the full

fields in Fig. S8–S12. As in Fig. 4, in Fig. 5 the "collapsed" sensitivity signals are referred to as "positive" ("negative") in the text if at least $90\,\%$ of the values in the sensitivity field are positive (negative). In Fig. 5a we see that Ta850 and 2T have the strongest influence on VO in the Rainy groups and by all non-anomaly temperature fields and TCWV in the Dry groups. In group Dry+Calm TCWV has the most control on VO. Sensitivities to temperature at all levels and moisture are mostly negative in all groups but Dry+Windy, in which they are positive.

For WS850 (Fig. 5b) the average sensitivities are generally similar as for VO but T850 has more control than Ta850 in group Rainy+Windy. Group Rainy+Calm does not have statistically significant sensitivity values from these two precursors but is instead controlled mostly by WS300 and T300. The prevalent signs and differences between intensity groups change little compared to VO sensitivities.

For WFP (Fig. 5c) the sensitivity values are quite similar to VO and WS850 with some minor differences. Group Dry+Windy

is controlled again the most by positive perturbations in T300 but now followed by WS300, with other contributions being much smaller. Groups Rainy+Windy and Dry+Calm have the largest sensitivities from temperature fields, with emphasis on the lower levels and the values being negative. Dry+Calm also has large negative sensitivity from TCWV. Group Rainy+Calm has the strongest control from GAMMA, followed by T850 (both negative).

As seen in Fig. 5d, sensitivity signals for SSI are mostly nonsignificant, which is likely due to the highly non-Gaussian

distribution of SSI. The only large values are in group Dry+Windy for which WS300 and T300 have the largest sensitivity values (both positive). Contrary to the other wind-based intensity measures, group Dry+Calm has positive sensitivity for SSI from temperature at all levels. These values are, however, small in magnitude. For group Rainy+Windy there are no significant sensitivity signals and for group Rainy+Calm the only significant ones are signals from PVa300.





Precipitation (Fig. 5e) has the most consistent sensitivity signals between groups. All groups exhibit the same positive and
strongest signals from temperature at all levels and TCWV. Like for VO and WS850, the strongest control in group Dry+Calm
for PRECIP is from TCWV. However, unlike for VO and WS850 (and WFP), the signals are overwhelmingly positive. The
change in sign in the sensitivity signal from negative for VO, WS850, and WFP to positive for PRECIP is true for group
Rainy+Windy as well. This indicates that from a statistical perspective, compound effects in ETCs in group Rainy+Windy
are not the most common case, and that wind speed and precipitation are anti-correlated. Based on a visual inspection of the
sPCA feature space in Fig. 1, this seems to be true and in fact is, however, weakly (Pearson correlation coefficient of $-0.21$).
In group Dry+Calm the anti-correlation is almost non-existent (Pearson correlation coefficient of $-0.06$). This relationship
between wind and precipitation sensitivities is also evident in the opposing signs of sensitivities to WS300 for PRECIP and
the wind-based measures in group Rainy+Windy. However, it should be noted that by definition all values of wind speed and
precipitation are large or small in groups Rainy+Windy and Dry+Calm, respectively. Group Rainy+Calm has clearly the largest
sensitivity signals from T300 whereas for group Dry+Windy the contribution is more evenly distributed between precursors.

Many of the sensitivity signals and patterns in the intensity groups do not have evident explanations based on theoretical
expectations. However, some of them can be understood via climatological conditions. In an individual intensity group the
genesis conditions are more homogenous than in the full dataset, which means that the caveats regarding the attribution of
sensitivity patterns to climatology discussed in Sect. 3.1 become less relevant. Climatological conditions affecting the genesis
environment could explain, for example, why wind intensity in group Dry+Windy is so sensitive to T300. Group Dry+Windy
has plenty of ETCs along the full extent of the main storm track (Fig. 2d) and especially the most varied genesis locations
(Fig. S2). ETCs closer to the start of the storm track are on average more intense than ones closer to the end (Dacre and Gray,
2013; Binder and Wernli, 2025; Cornér et al., 2025). Like the average intensity, the climatological T300 field has decreasing
values along the storm track (Fig. S4), hence the strong sensitivity to T300 in group Dry+Windy. Similarly, wind intensity in
group Dry+Calm is negatively sensitive to temperature and TCWV. This indicates that ETCs whose genesis is more northerly,
specifically in this group outside the Mediterranean, have stronger winds (Fig. 2c). The reason why PRECIP in group Dry+Calm
is especially sensitive to TCWV is the same, with the added effect of limited moisture availability in the northern parts of
the domain (Pfahl and Sprenger, 2016). In these areas relatively small changes in TCWV would thus lead to much more
precipitation.

## 405    4   Differences between ETC intensity groups and their temporal evolution

Answering the question of how the genesis environment of an ETC affects its intensity later in the life cycle in general can
be done based on the results shown in Sect. 3.1. This is supported by the genesis composites of the precursor fields in the
four intensity groups shown in Fig. S8. For example, Rainy groups have warmer and more humid genesis environments while
Windy groups have stronger upper-level jets and baroclinicity. Thus, there is a clear physical relationship between the genesis
environment of an ETC and the resulting intensity in the full dataset.





**Figure 5.** Collapsed sensitivity values for the five intensity measures in the four groups. Missing values (white squares) indicate that all signals in the 2D sensitivity field are nonsignificant. The plus (minus) sign in front of a number indicates that at least 90 % of the significant sensitivity values are positive (negative). The labels on the y-axis refer to the groups by using the initials R: Rainy, C: Calm, W: Windy, and D: Dry.

In Sect. 3.2 we showed how the sensitivity effects appear in the intensity groups. While there are differences between the sensitivity signals for the intensity groups, none of the groups exhibits unique behaviour in its sensitivity patterns. For example, the PRECIP sensitivities are very similar among the groups, and the wind-based measures have qualitatively similar behaviour for the two most different groups, Rainy+Windy and Dry+Calm. Based on these results, none of the groups is distinct in terms
of what controls the intensity and how. They are, however, different enough among each other and compared to the full dataset that we cannot say that ETC intensity is controlled by the same precursors in the same manner for all ETCs regardless of their intensity, apart from PRECIP perhaps.





Possible explanations for the observed non-distinct patterns include 1. the sensitivity does not meaningfully differ for ETCs of different intensities, 2. the sensitivity fields represent climatological shifts in the genesis environment instead of dynamically perturbed background states, and 3. the ESA method does not capture non-linear effects occurring later in the ETC life cycle which cause differences in intensity. Of these the most likely and most insightful explanation is the latter, which is supported by the fact that when we try to answer the question of "how much" perturbations in the genesis environment affect the intensity, the answer is "relatively little". Neither in the full dataset nor the intensity groups are the magnitudes of the sensitivity values large given the variability in intensity (not shown). For all intensity measures the largest sensitivity values are much smaller than both the standard deviation in the full dataset and the differences in the average values of the intensity groups. Even with a perturbation of multiple standard deviations in magnitude in a precursor field, the resulting change does not satisfactorily explain differences in intensity. These results together with those obtained by Graf et al. (2017) indicate that variability in ETC genesis states does not clearly correspond to variability in ETC intensity. Graf et al. (2017) analysed 30 ETC precursors in the northern hemisphere and performed principal component analysis on them to obtain an objective ETC cyclogenesis classification. Their four cyclogenesis classes clearly differed in their genesis composites and they were able to link them to the ETC life cycle composites of Petterssen and Smebye (1971). Furthermore, their analysis showed that upper-level potential vorticity is an important feature at differentiating between cyclogenesis states, whereas in our results it has little association with the variability in intensity.

The overall small sensitivity values might also explain why the sensitivity patterns for the four wind-based intensity measures are very similar (Fig. 3). The ESA is able to capture only the linear part of the wind variability, whereas the differences among the wind-based intensity measures arise from non-linear processes, especially in the case of SSI (Cornér et al., 2025). These observations do not, however, mean that the ESA method is unsuitable to identify controls of ETC intensity at genesis time. This is showcased by the physically meaningful sensitivity patterns seen in the full dataset. Instead, it simply means that we need to investigate processes after genesis to understand differences in ETC intensity better. To study these non-linear processes present in the ETC intensification which could explain the differences, we investigate the evolution of both the intensity measures and the precursors in the following.

Before investigating the time evolution of the ETC intensity measures in the different groups, we first describe the general features of their distributions at time of maximum intensity from the sPCA split. The sPCA grouping efficiently reduces overlap in the intensity measures present in the first two PCs, i.e. wind speeds and precipitation. It does less so for e.g. WFP, VO, and MSLPa which are present mostly or only in PC3 and PC4. Moreover, the sPCA grouping splits the intensity measure distributions in a binary fashion, i.e. groups Rainy+Windy and Rainy+Calm have very little overlap in wind speed but a lot in precipitation, whereas groups Rainy+Windy and Dry+Windy have very little overlap in precipitation but a lot in wind speed. This is visible in the distributions at the time of maximum VO in Fig. 6.

## 4.1 Intensity measure evolution

The life cycles between $-96\,\mathrm{h}$ and $96\,\mathrm{h}$ from the time of maximum VO of six selected intensity measures in the intensity groups are shown in Fig. 6. The selected intensity measures include four already discussed ones (VO, WS850, PRECIP, and



WFP) as well as two additional ones – mean sea level pressure anomaly (MSLPa) and maximum 10 m wind gust (FG10). Details of their calculation can be found in Cornér et al. (2025). They were included in the analysis due to their different behaviour, offering additional insight into the differences in intensification among the groups. As described above, for each
intensity measure the groups fall into two rough regimes either based on the Dry–Rainy axis or the Calm–Windy axis, i.e. either the Windy groups are in one regime and the Calm ones in the other, or the Rainy groups are in one regime and the Dry ones in the other, depending on the intensity measure. For most intensity measures the regimes are evident from around 72 h before the time of maximum VO until the end of the investigated time period (96 h after), and the difference between the regimes peaks simultaneously with the intensity. This indicates that on average ETCs in either of the two more intense groups (i.e. Windy
or Rainy) undergo strong non-linear intensification. In fact, ETCs in the more intense groups see at least a doubling of the wind-based intensity measure values during the investigated period. When analysing the relative magnitudes of the intensity measures towards the ends of the investigated period, it should be acknowledged that the small sample size and its variations among the groups may introduce a bias.

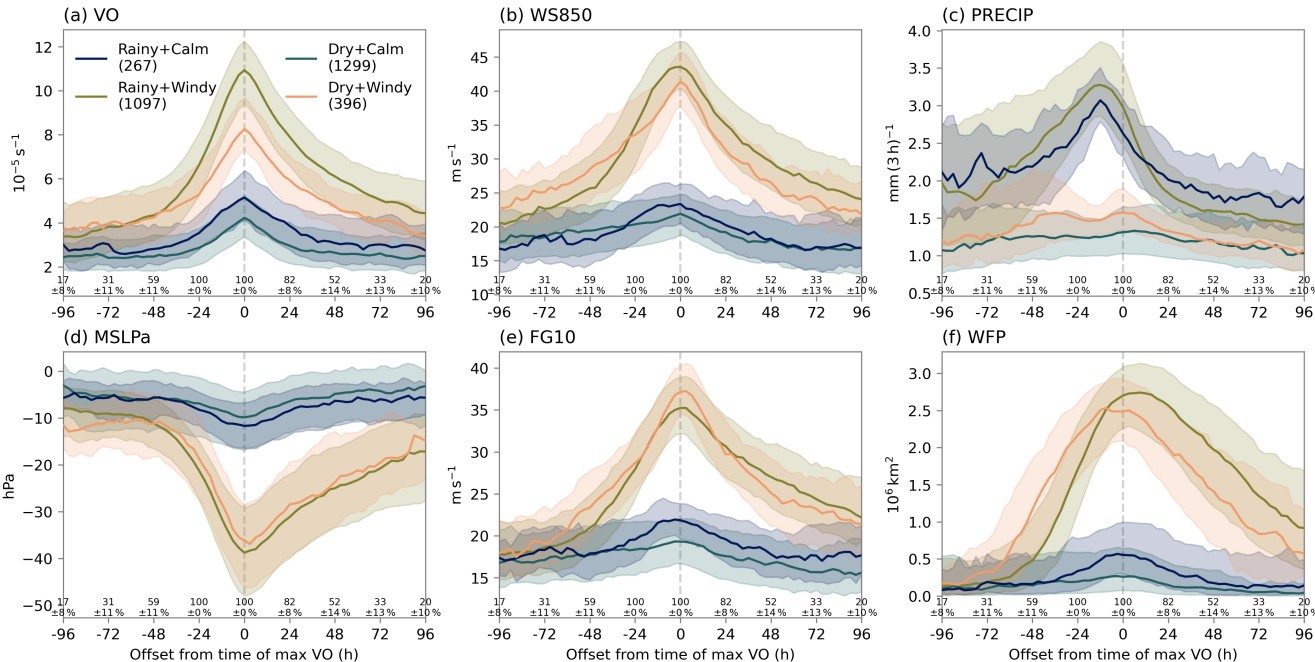

**Figure 6.** Average evolution of six selected intensity measures in the four intensity groups relative to the time of maximum VO. The solid line shows the median value and the shading contains values in the two middle quartiles. Percentages on the x-axis show the mean and standard deviation of proportion of tracks sampled in the groups for a given time step. The complete group sizes are shown in parentheses in the legend in panel (**a**). Median times of maximum VO in the groups are shown in Fig. S13.

As described above, overlap between the VO distributions is not reduced only by the Windy–Calm split but also be-
tween groups Dry+Windy and Rainy+Windy, with the latter reaching the highest values (Fig. 6a). However, the overlap be-



tween Windy and Calm groups is greatly reduced only between around $-12\,\mathrm{h}$ and $+24\,\mathrm{h}$ when the 25th percentile of group Dry+Windy is larger than the 75th percentile of Rainy+Calm. In addition, VO values in group Rainy+Windy are on average lower than in group Dry+Windy until $48\,\mathrm{h}$ before time of maximum VO. Together these factors indicate that in group Rainy+Windy VO increases very rapidly, especially in the $24\,\mathrm{h}$ before maximum VO. The decrease in VO is almost as fast as
the increase. The two Calm groups exhibit similar symmetric growth and decay as the Windy ones but much milder.

Group Rainy+Windy exceeds Dry+Windy in MSLPa at the same time as in VO ($-48\,\mathrm{h}$) but after this mark, their evolution closely follows the same values (Fig. 6d). Unlike for VO, Windy ETCs weaken in MSLPa slower than they intensify. For WS850 Rainy+Windy ETCs reach on average larger values than Dry+Windy only after around $-24\,\mathrm{h}$. After this, however, the values are on average constantly larger (Fig. 6b). An interesting contrast to this is FG10 in which the values are on average
larger in Dry+Windy ETCs before and at time of maximum VO, all the way up to $+24\,\mathrm{h}$ after which Rainy+Windy becomes larger (Fig. 6e). This is likely due to the atmosphere being drier, which allows more dry-adiabatic vertical profiles and thus more momentum carried into the boundary layer. In WFP this is seen as a phase shift between the Windy groups (Fig. 6f). The Dry+Windy ETCs lead the Rainy+Windy ones by around $24\,\mathrm{h}$, with the latter reaching a slightly higher peak around $+12\,\mathrm{h}$. The phase shift can also partly be attributed to differences in the track density distributions of the two groups (Fig. 2b,
d). Dry+Windy reach on average larger WFP values earlier because of their more marine locations in which friction is not inhibiting strong wind gusts as much as over land.

The evolution of PRECIP in the intensity groups differs from the wind-based intensity measures (Fig. 6c). Differences in intensity between the groups, now split in the Dry–Rainy axis, are more evident throughout the investigated period. Of the Rainy groups, Rainy+Windy has larger values than Rainy+Calm on average between $-60\,\mathrm{h}$ and $+12\,\mathrm{h}$, indicating that
it experiences both more rapid growth and decline. Both of them reach their peak around $-12\,\mathrm{h}$, which is consistent with previous research (e.g. Pfahl and Sprenger, 2016; Cornér et al., 2025). Especially for the Rainy ETCs, the range of values is large initially but reduces drastically around the time of maximum VO. The range increases again in group Rainy+Calm but not as much in Rainy+Windy. The reason why group Rainy+Calm has a smoother PRECIP evolution than Rainy+Windy could be related to moisture availability which is more consistent for the former group due to its ETCs perpetually existing in the
southern parts of the domain. The Dry groups undergo modest or almost no increase in PRECIP values during their evolution.

The results from this analysis indicate that while group Rainy+Windy has on average the most intense ETCs in terms of both wind and precipitation, it also undergoes the most rapid intensification in both of these aspects. This is consistent with e.g. Cornér et al. (2025) who showed that the most intense ETCs also have the largest deepening rates. Moreover, group Rainy+Calm is more intense and intensifies faster than Dry+Calm in the wind-based measures. The reason for the Rainy
groups experiencing stronger wind intensification than their Dry counterparts may at least partly be due to the diabatic heating from precipitation further intensifying the ETC and delaying the time of maximum VO and wind speed. The wind intensity, especially VO, can in turn affect the precipitation intensity. ETCs with similar moisture availability have more precipitation if they have stronger vorticity values and therefore more convergence. Via continuity this induces stronger vertical motion and more precipitation (Field and Wood, 2007). This effect could partly explain why group Dry+Windy has larger PRECIP values





than group Dry+Calm (Fig. 6c). The diabatic heating effect is likely more important in the Rainy groups in which PRECIP peaks before VO and VO peaks later than in their Dry counterparts (Fig. S13), but in the Dry groups it is less clear.

By examining the evolution of six intensity measures, we showed that the growth of ETC intensity is in some cases highly non-linear but also notably variable between ETCs of different maximum intensities. In the following we investigate the evolution of the precursors leading up to the time of maximum VO to identify processes influencing the intensification and explaining the differences which are not evident from the ESA.

### 4.2 Precursor composite evolution

The evolution of WS300 in the intensity groups is shown in Fig. 7 as composites. In WS300 we see clear differences in the composites already at genesis time, which is in agreement with the results from the ESA in Fig. 3. The Windy groups have much stronger jets and the ETC centres are located in the right entrance of a jet streak (Fig. 7b, d), a favourable location for cyclogenesis. These groups also increase noticeably in WS300 during ETC development unlike the Calm groups. The WS300 values at time of maximum VO are slightly larger in Rainy+Windy than Dry+Windy, and the flow is more tilted towards the northeast relative to the travel direction as well as more amplified, with large WS300 values also northeast of ETC centre downstream of the main jet streak maximum (Fig. 7v, x). ETCs in group Rainy+Windy have also travelled across the jet streak to the left exit later than in group Dry+Windy ($-12\,\mathrm{h}$ compared to $-24\,\mathrm{h}$). This is consistent with results shown in Sect. 4.1 which indicate that ETCs in group Rainy+Windy undergo longer and/or later intensification. Noteworthy is also that the jet is at its strongest at $-12\,\mathrm{h}$ instead of time of maximum VO. This makes sense, since after time of maximum VO, ETCs begin to weaken. An interesting feature is also the fact that in the Rainy groups the central MSLP value drops notably between $-12\,\mathrm{h}$ and time of maximum VO (by about $4\,\mathrm{hPa}$). This does not occur in the Dry groups which reach their minimum MSLP at $-12\,\mathrm{h}$. This is consistent with results presented in Sect. 4.1. The Rainy groups have stronger lower-level PV anomalies than their Dry counterparts at $-12\,\mathrm{h}$ and at time of maximum VO (not shown). This indicates that the reason for the differences in late intensification are due to diabatically produced PV, as hypothesised in Sect. 4.1.

Unlike WS300, PVa300 does not show large differences between the groups at genesis time (Fig. 8a–d). This is what the ESA indicates as well with small PVa300 sensitivities, i.e. PVa300 field at genesis time has little effect on the intensity of the ETC (Fig. 3 and 4). However, differences in the PVa300 fields arise during ETC development. In the Windy groups at $-48\,\mathrm{h}$ a positive PV anomaly wraps cyclonically from the northeast to the west and a negative PV anomaly from the southwest to the east around the centre of the ETC (Fig. 8j, l). Stanković et al. (2024) showed that in cases with extreme surface winds the positive upper-level PV anomaly stretching northeast of the developing ETC centre is associated with a pre-existing ETC downstream. A cyclonic flow around the pre-existing ETC advects the positive PV anomalies southward towards the developing ETC centre. For both Windy groups the positive PV anomaly grows and approaches the ETC centre as the ETC intensifies, but the negative anomaly grows strong only in Rainy+Windy (to about half the magnitude of the positive anomaly). Moreover, in group Rainy+Windy the eventual negative PV anomaly extends well into the eastern and northeastern sector of the ETC, whereas in group Dry+Windy it recedes, resulting in the strongest gradient being in the meridional direction. The strong zonal gradient at time of maximum VO in group Rainy+Windy (Fig. 8v), which may be amplified by a strong warm conveyor belt





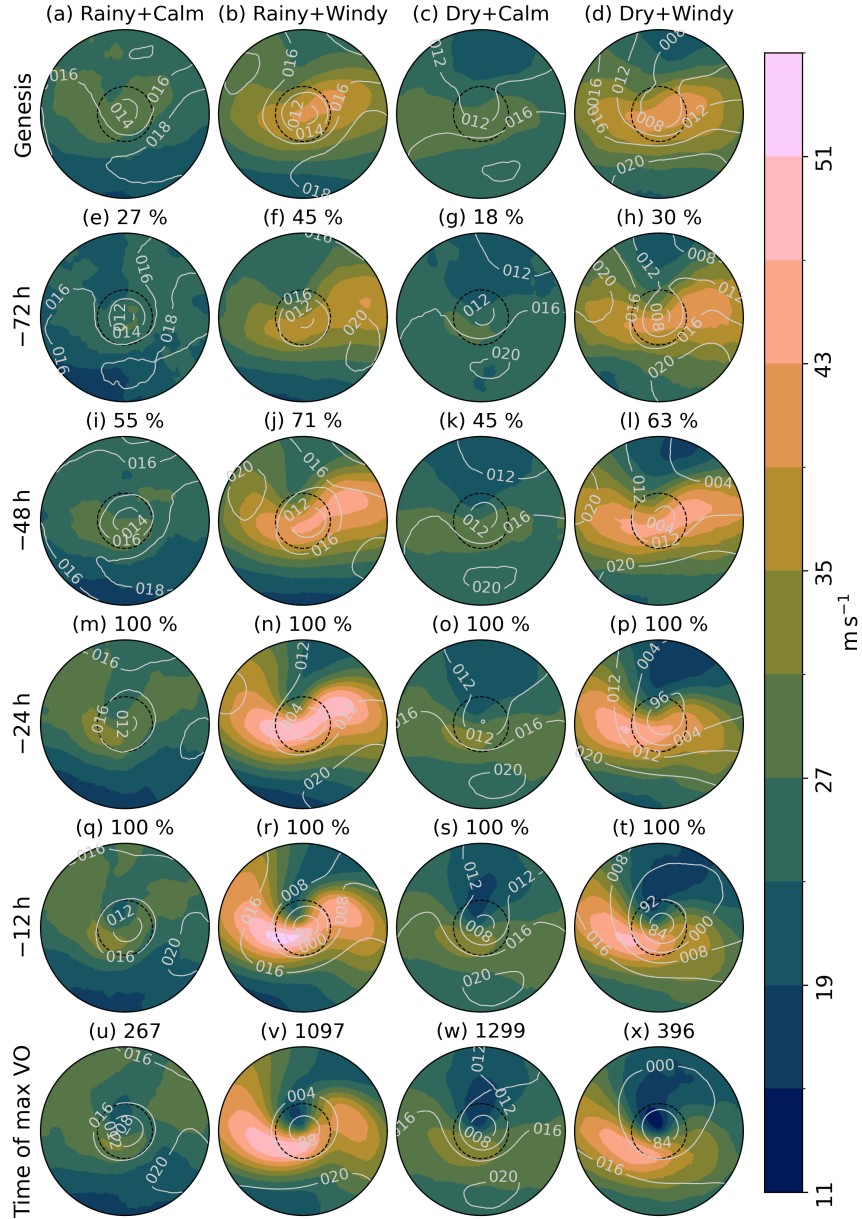

**Figure 7.** Composite evolution of WS300 (colours) in the four intensity groups. MSLP composite is shown in the contours in hPa with the leading number (1 or 9) omitted. The dashed black circle denotes a radius of $6°$ geodesic and the radius of the whole circle is $18°$ geodesic. The numbers on top of panels in the last row indicate the total size of the group. Percentages on other rows indicate the proportion available to sample of these sizes. Genesis time includes full sample, the same as the time of maximum VO.

outflow creating negative upper-level PV anomalies (Schemm and Wernli, 2014; Madonna et al., 2014), is consistent with the



WS300 distribution with a slightly amplified flow and strong winds also close to the upper-level ridge, downstream of the
main jet streak (Fig. 7v). Groups with Calm winds develop positive PVa300 values close to the ETC centre which are weak
in magnitude, consistent with the weak WS300 values. The positive PV anomalies are located close to the centre throughout
the evolution, which implies that these ETCs are more vertically stacked, indicating weaker cyclogenesis from baroclinic
instability (Hoskins et al., 1985). Čampa and Wernli (2012) also found that more intense ETCs in terms of MSLP have clear

intensification of PV anomalies during the $24\,\mathrm{h}$ before time of maximum intensity. Thus, the more intense ETCs also have
stronger PV anomalies associated with them.

If PVa300 showed little differences between groups and WS300 only on the Calm–Windy axis, the T850 distribution is
distinct for each group already at genesis time (Fig. 9a–d). Rainy groups have on average higher temperatures while Windy
groups have larger temperature gradients. The differences in T850 gradients arise from colder air on the northern side of the

ETC centre when comparing groups with similar precipitation values. Qualitatively the distributions match the expectations
from the ESA: warmer background environment is associated with more precipitation and stronger baroclinicity with stronger
winds (Fig. 3). All groups have a hint of a thermal wave already at genesis but only in the Windy groups does it grow notably
in amplitude, revealing a clear warm sector extending southwest of the ETC centre at time of maximum VO. The groups have
the strongest T850 gradients at different times during the life cycle. This may be related to differences in genesis time relative

to the time of maximum VO or how quickly on average the ETCs traverse the jet as discussed in relation to Fig. 7. Only the
composite of group Rainy+Windy develops a warm front which is nearly perpendicular to the cold front and comparable in
strength (Fig. 9v). At upper levels, T300 shows similar features as T850 (not shown). Differences include smaller temperature
gradients and the Windy groups developing a seclusion of warm air at time of maximum VO which is likely due to the $300\,\mathrm{hPa}$
surface dipping low into the troposphere.

The composite evolution of TCWV is shown in Fig. 10. The distributions at genesis time resemble qualitatively those of
T850. The Rainy groups have larger values while the Windy groups have larger meridional gradients (Fig. 10a–d). However, the
evolution differs from that of T850. The Dry groups see little change in the TCWV distribution whereas the Rainy ones evolve
to exhibit patterns indicating large moisture values in the warm sector of the ETC. The maximum TCWV values coincide with
the maximum in PRECIP at $-12\,\mathrm{h}$ during which the Rainy groups show the warm conveyor belt air stream, with especially

group Rainy+Windy having a narrow band of large TCWV values extending from the southwest to the centre of the ETC
perpendicular to the warm front (Fig. 10q, r).

The GAMMA composite evolution shown in Fig. 11 is slightly noisy but shows nonetheless interesting and distinct features.
Qualitatively, the results agree with those of the ESA. The Windy intensity groups have slightly smaller values (a more stable
atmosphere) at genesis than the Calm groups (Fig. 11a–d). The GAMMA values do not, however, change much through time in

the Calm groups. They only slightly increase close to the ETC centre. During the ETC evolution the Windy groups experience
changes in their GAMMA distribution which align with the stronger growth in intensity and possibly explain why the genesis
sensitivities from GAMMA do not match theoretical expectations. Group Dry+Windy has the largest increase in GAMMA
values and ultimately the steepest lapse rates of which the maximum values are close to the ETC centre at time of maximum
VO (Fig. 11x). The large values indicate close to dry-adiabatic lapse rates. Although GAMMA is calculated between 850





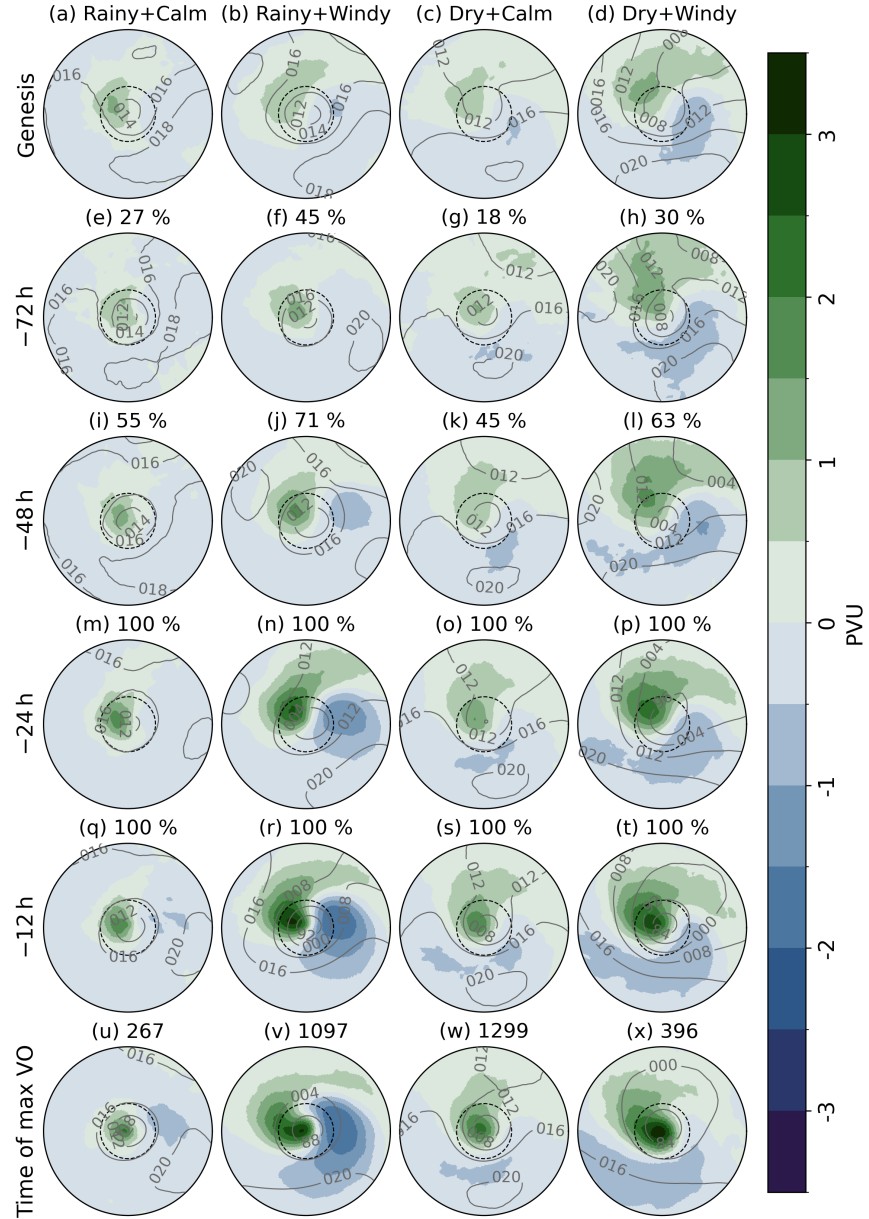

**Figure 8.** As in Fig. 7, with the shading showing PVa300.

and $500\,\mathrm{hPa}$, the large GAMMA values imply steep lapse rates closer to the surface as well, thus enabling high-momentum air to cause strong winds near the surface (Pantillon et al., 2018). Strong wind gusts have been shown to be associated with evaporative cooling for which a dry atmosphere is a prerequisite (Browning et al., 2015; Tam et al., 2025). This could partly explain the difference of relative magnitudes in FG10 and WS850 between groups Rainy+Windy and Dry+Windy shown




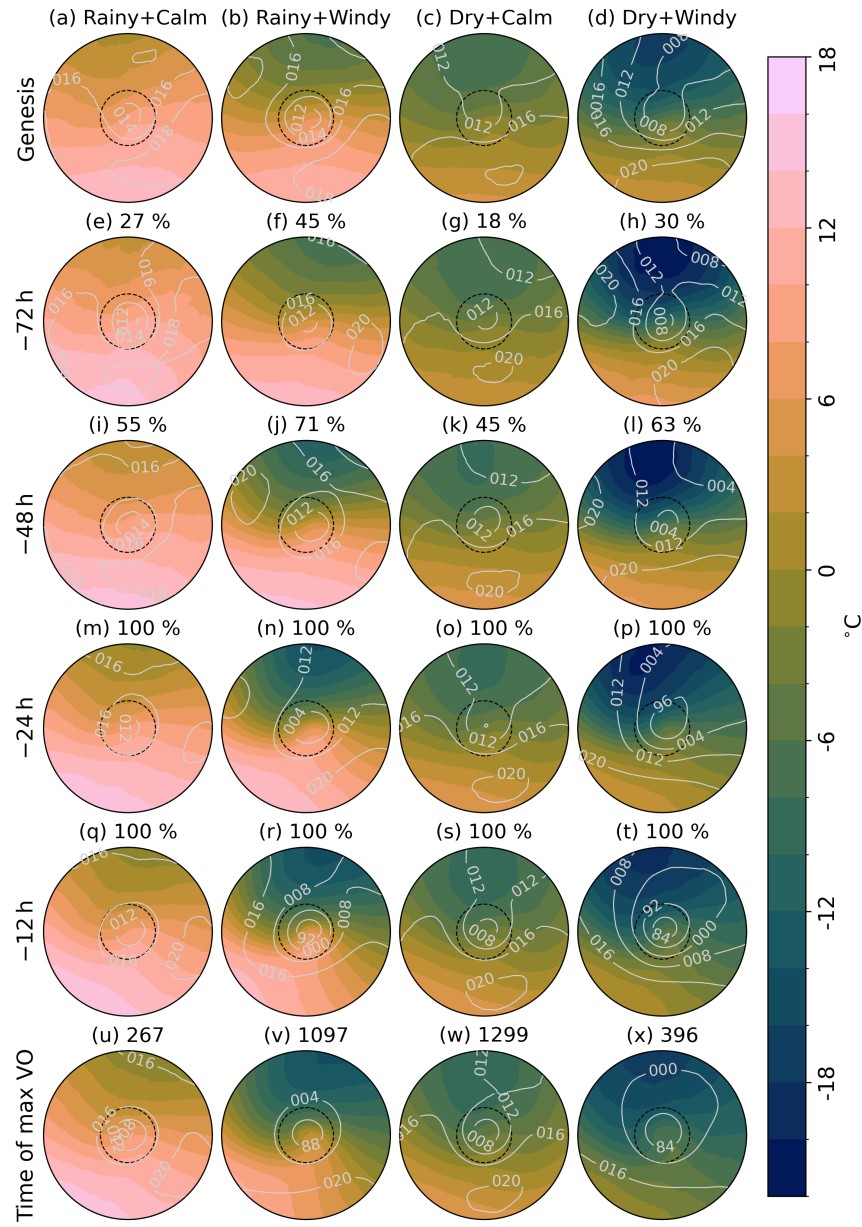

**Figure 9.** As in Fig. 7, with the shading showing T850.

in Fig. 6, i.e. why group Rainy+Windy has larger WS850 values but smaller FG10 values than group Dry+Windy. Group
Rainy+Windy develops the largest GAMMA values in the centre of the ETC and in the warm sector. Behind the warm front it
has small GAMMA values which aligns with the vertical profile of warm air on top of colder air (Fig. 11v).





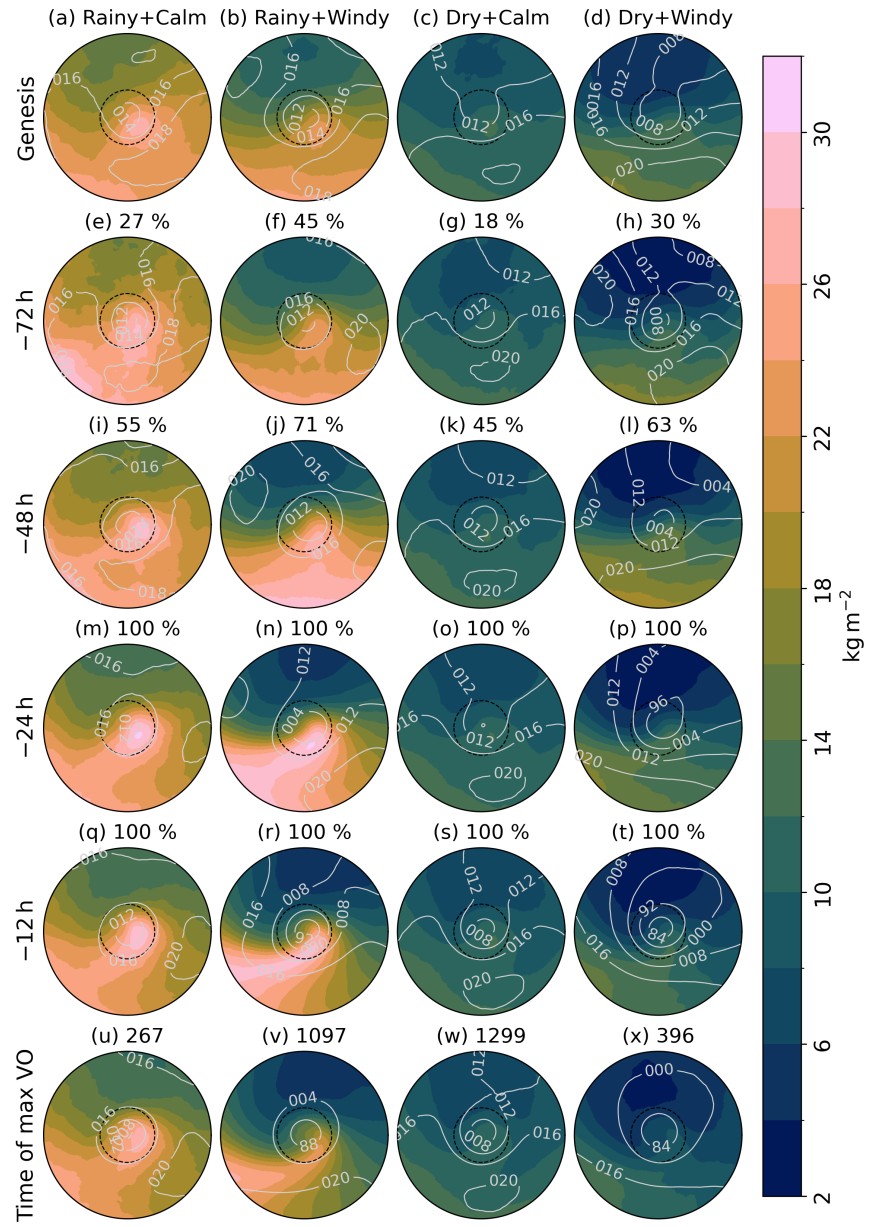

**Figure 10.** As in Fig. 7, with the shading showing TCWV.

In summary, differences in the intensity evolution among the intensity groups are due to both dynamical (e.g. WS300) and thermodynamical (e.g. TCWV) differences in the precursor fields and their evolution. These results highlight the importance of investigating multiple different precursor fields when trying to understand the details of differences in ETC intensity and intensification. That being said, the largest differences in the precursor composites begin to be evident between around 48 h and



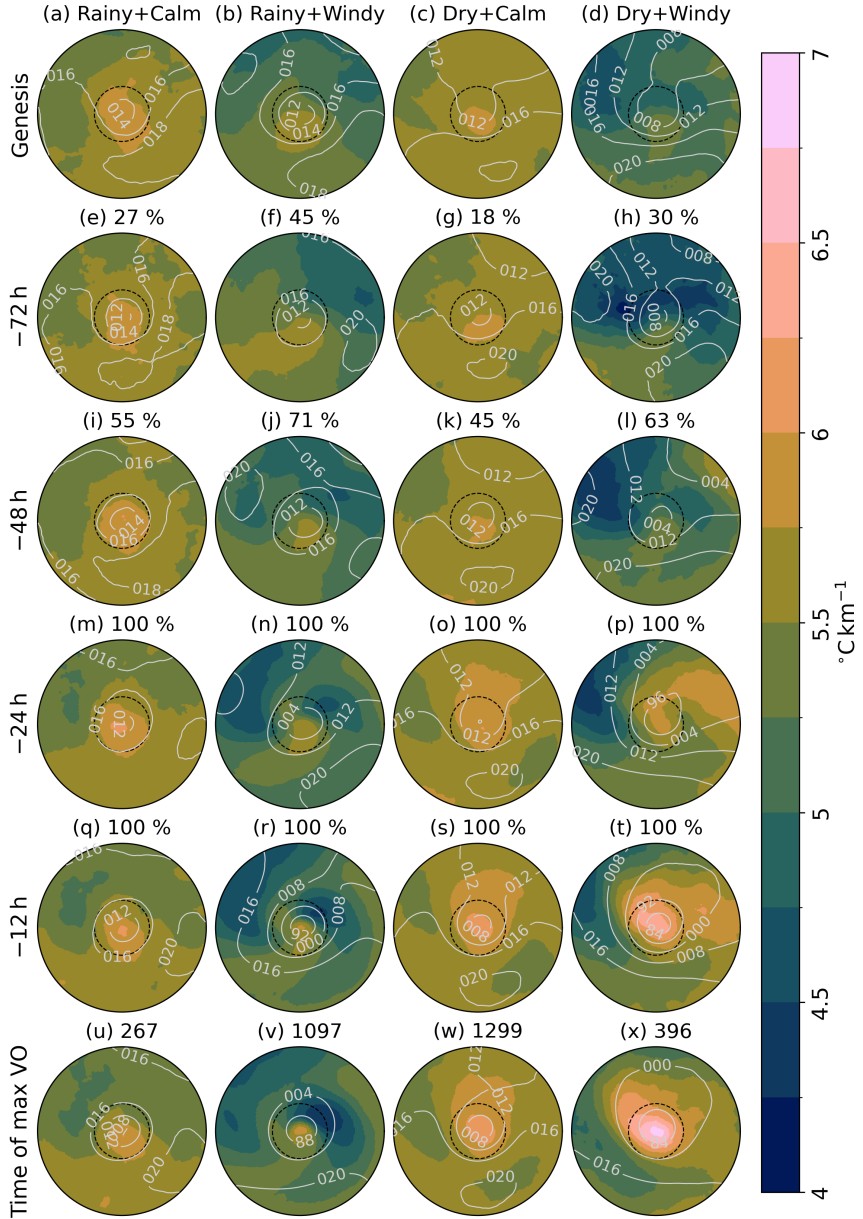

**Figure 11.** As in Fig. 7, with the shading showing GAMMA, defined as $-[T(850\,\mathrm{hPa}) - T(500\,\mathrm{hPa})]/[Z(850\,\mathrm{hPa}) - Z(500\,\mathrm{hPa})]$, where $T$ is temperature and $Z$ geopotential height.

24 h before the time of maximum VO. This coincides with the time when most intensity measures start to experience non-linear growth (Fig. 6). However, there are clear differences in average intensity already before this between the groups. Based on this it is, therefore, possible to at least qualitatively attribute the changes in intensity to changes in the precursor fields.



## 5 Conclusions

In this study we investigated the links between extratropical cyclone (ETC) intensity and precursors in the North Atlantic and Europe in the extended cold season. Specifically, we used a dataset spanning 43 extended winters of ETC tracks to quantify how much the state of multiple ETC precursors at time of genesis influences the eventual maximum intensity later on in the ETC life cycle. The relationships between 9 ETC precursors and 5 intensity measures were quantified using a method called ensemble sensitivity analysis (ESA). ESA is a statistical method that quantifies the relationship between two variables based

on linear regression from a sample (ensemble) of features. The results of the ESA indicated that the wind intensity of an ETC is controlled mostly by wind speed and temperature at upper levels ($300\,\mathrm{hPa}$) as well as the temperature gradient at lower levels ($850\,\mathrm{hPa}$), and the mid-tropospheric lapse rate. Specifically, ETCs were found to be more intense in terms of wind when at genesis time the jet is stronger downstream of the ETC centre and the temperature gradient at upper levels is stronger with an emphasis on warming on the southern side. An increased lower-level temperature gradient and reduced lapse rate were

also associated with stronger winds in ETCs. More precipitation was associated with warmer temperatures and more moisture throughout the troposphere. Potential vorticity anomalies at upper levels at genesis time were found to have little control on both wind intensity or precipitation.

The ESA was repeated for ETC groups of different intensities to determine if the relationship between genesis conditions and the eventual cyclone intensity varies with the type of ETC. The intensity groups were extracted from a sparse principal

component analysis (sPCA) performed on the set of ETC tracks and 11 associated intensity measures from Cornér et al. (2025). The sPCA feature space had wind speed and precipitation as dominant features in its two first components, respectively. The feature space was split in these two directions and average-intensity ETCs were removed, resulting in four distinct intensity groups called Rainy+Windy, Rainy+Calm, Dry+Calm, and Dry+Windy. Some differences in the results of the ESA were found between the groups but these were not entirely unique, leading to the conclusion that the genesis environment does not explain

differences in ETC intensity.

While the observed sensitivity patterns in the full dataset can be explained by theories on cyclogenesis and ETC intensification, the magnitude of the sensitivity signals was small compared to the variability in intensity. Given this, and the relatively small differences between the sensitivity values in the intensity groups, we conclude that in quantitative terms the genesis state has only a small effect on the eventual intensity of an ETC. The results obtained in this study can, however, be qualitatively used

to estimate how the intensity of ETCs will change in the future climate. It has been shown that ETC-associated precipitation is likely to increase in the future (Catto et al., 2019). Our results support this by associating a warmer and more humid atmosphere with more precipitation. The effect on wind intensity is less clear since upper-level baroclinicity is projected to increase, lower-level baroclinicity to decrease, and static stability to increase (Catto et al., 2019). Based on our results the ETC wind footprint, for which stability has the most control, might grow in the future. For relative vorticity and wind speed at $850\,\mathrm{hPa}$

the effect would depend on the relative magnitude of the changes in the precursors. Increasing stability would act to increase the two intensity measures while decreasing lower-level baroclinicity would decrease them. Due to the large importance of





the upper-levels based on our results, the increasing upper-level baroclinicity along with warming would likely tip the scales towards more intense winds and stronger vorticity.

To better understand the differences in ETC intensity, we investigated the temporal evolution from genesis time to time of maximum intensity of both the intensity measures as well as the precursors in the groups. The precursors were analysed around the ETC centre using cyclone-relative composites. Like the magnitudes of the intensity measures themselves, their evolution was clearly split through the Rainy to Dry and Windy to Calm divide. In almost all intensity measures the ETCs in group Rainy+Windy undergo the most rapid intensification and reach on average the largest values. An exception to this is 10 m wind gust for which group Dry+Windy reaches the highest values, likely because of steeper lapse rates due to a drier troposphere allowing momentum transport to the surface. The larger wind intensity of the Rainy groups compared to their Dry counterparts is due to either intensifying diabatic effects or stronger vertical motion in the more intense ETCs causing larger precipitation values. It is also possible that the observed relative intensities are a combination of these two effects due to a feedback mechanism (Sinclair and Catto, 2023). The groups with the larger values of wind speed and precipitation, i.e. Windy and/or Rainy, at least double in their intensity during the $96\,\mathrm{h}$ before time of maximum intensity. The observed differences in growth of intensity and the fact that the intensity measure values differ between the groups the most at time of maximum intensity indicate that some ETCs experience non-linear intensification. The non-linear growth explains why ESA, a linear method, is unable to quantitatively represent the factors leading to the observed variability in ETC intensity. This is the main limitation of the method. It does not, however, reduce the importance of the results when interpreted as climatological relationships. Another limitation related to our application of ESA is the point of genesis which is dependent on the ETC tracking method used. This is, however, not a major issue since our cyclogenesis regions generally agree with previous studies using the same tracking method with different data (e.g. Dacre and Gray, 2009; Hodges et al., 2011; Priestley et al., 2020) and different tracking methods (e.g. Pinto et al., 2005; Wernli and Schwierz, 2006).

Consistent with the results of the ESA, the precursors determined to be the most important for ETC intensity showed differences in composites between the intensity groups already at genesis time. For example, Windy groups had stronger jets and larger temperature gradients, while Rainy groups were more humid and warmer. Unique combinations of these features were thus seen in the temperature composites at genesis. The large differences in intensity between the groups become more evident when investigating the evolution of the precursors fields. The differences seen in the genesis composites grow larger and more unique as the ETC intensifies. Moreover, precursors which do not have differences at genesis and to which the intensity measures are not sensitive, namely the potential vorticity anomaly at $300\,\mathrm{hPa}$, exhibit clear variability in their evolution. In most cases the differences in the precursors arise most clearly between $48\,\mathrm{h}$ and $24\,\mathrm{h}$ before the time of maximum vorticity, which is consistent with differences seen in the evolution of the intensity measures and their non-linear growth. This also explains why in studies which calculated sensitivities for these kind of time offsets (Dacre and Gray, 2013; Laurila et al., 2021) the sensitivity values are generally larger than here. Differences in average intensity between the groups can, however, be seen already before this. This means that differences in intensification can be attributed to, but not necessarily easily predicted from, the precursor fields.

Features seen in the precursor evolution include, for example:



- a clear traversing of the jet streak in the Windy groups and little development in upper-level wind speed in the Calm groups

- development of upper-level potential vorticity anomalies close to the ETC centre with larger values in the Windy groups

- development of frontal structures and air mass sectors, with the strongest temperature gradients and clearest fronts in group Rainy+Windy

- large total column water vapour values in the warm sector along narrow filaments indicative of moisture transport in a warm conveyor belt in the Rainy groups and little change in moisture in the Dry groups

Some of the observed sensitivity patterns and differences between the intensity groups in their precursor evolution can be
attributed to climatological conditions mandated by geography. Generally, ETCs in the Windy groups occur along the main North Atlantic storm track, an area determined by the climatological location of the jet stream. Calm ETCs occur mostly over land, in the Mediterranean, or at the edges of the main storm track. ETCs in the Rainy groups occur closer to the southern parts of the North Atlantic and Europe. However, the Mediterranean are has some Dry+Calm ETCs as well. Despite the on average different genesis and occurrence locations of the intensity groups, there is considerable overlap between these regions.
Moreover, the variability in the precursor fields in a given group is roughly as large as differences between the precursor composites of the groups. Therefore, it can be estimated that the climatological distribution explains about a half of the typical variability in a precursor at genesis time. The remaining variability comes from local climatological temporal variability. While the variability in the precursors at genesis time explains some of the variability in ETC intensity, to fully understand observed differences in ETC intensification and maximum intensity, we need to investigate a combination of precursor fields and their
temporal evolution.

*Code and data availability.* ERA5 reanalysis data were downloaded from the Copernicus Climate Change Service (Hersbach et al., 2017). The extratropical cyclone track dataset and associated intensity measures are available at Cornér et al. (2024). The Python code and data used in the analysis are available at Cornér et al. (2025a) and Cornér et al. (2025b), respectively.

*Author contributions.* All authors contributed to the design of the study. JC performed the data analysis and visualization. All authors
contributed to the interpretation of the results. JC wrote the first draft of the manuscript and all authors reviewed and edited the manuscript.

*Competing interests.* The authors declare that they have no competing interests.



*Acknowledgements.* We wish to thank Kevin Hodges for providing the cyclone tracking software TRACK. We acknowledge CSC – IT Centre for Science, Finland, for computational resources and ECMWF for producing ERA5 reanalysis. This research was supported by the Research Council of Finland (grants no. 338615 and no. 368683). JC was funded by the University of Helsinki Doctoral School. This study uses scientific colour maps (Crameri, 2023) to prevent visual distortion of the data and exclusion of readers with colour vision deficiencies.




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
