# Peer review of "Identifying controls of extratropical cyclone intensity at genesis time and during intensification in the North Atlantic and Europe"

_EGUsphere, 2025_

## Referee Comment (RC1)

**Review of "Identifying controls of extratropical cyclone intensity at genesis time and during intensification in the North Atlantic and Europe" by Corner et al. 2025 submitted to WCD**

This study applies an ensemble-based sensitivity analysis to extended wintertime extratropical cyclones in the North Atlantic and Europe to identify precursor fields influencing maximum cyclone intensity using five different intensity measures. To quantify the controls on cyclone intensity, the authors apply the ensemble-based sensitivity analysis to multiple cyclone precursors at cyclone genesis time in a cyclone centered perspective. The study shows that the intensity of cyclones in terms of winds is controlled mostly by wind speed at upper levels and temperature gradients. Whereas cyclone intensity in terms of precipitation is mainly controlled by more moisture and warmer temperatures. As one finding of the study is that the genesis time has limited predictive value for cyclone intensity, the study additionally investigated the temporal evolution of the precursor fields.
The manuscript is clearly written and will be of interest to many readers of WCD. It will make a valuable contribution to the literature on cyclone intensification. I do, however, have some comments that should be addressed by the authors, mostly regarding the length and conciseness of the manuscript. Otherwise, I think this manuscript is suitable for publication in WCD.

**General comments**
1. While the manuscript is comprehensive, it is a bit lengthy. The readability could be enhanced by shortening and making it more concise to improve the overall flow. I would suggest to shift the focus more on the key findings. For example, but not limited to, Section 4 would benefit from greater conciseness and a clearer emphasis on the key new insights and main findings.

2. It is a bit misleading that wording cyclone 'intensity' is used in two ways, first different measures of cyclone intensity, based on dynamical and impact-based measures like VO, WS850, and PRECIP are introduced. Second, cyclone types are introduced by grouping in rainy and windy groups. It would help the readability to distinguish more clearly between those two types of cyclone intensity and what the difference between those two types of cyclone intensity is.
In section 3.2 of the manuscript, the sensitivity of the genesis precursors is compared to different cyclone intensity measures for different cyclone groups (windy and rainy). In Section 4.1, the evolution of cyclone intensity based on different intensity measures is compared across the different cyclone groups. However, it seems somewhat naturally that cyclone types associated with precipitation exhibit higher intensity when precipitation based measures are used, as this is somewhat true by definition. Similarly, intensity measures related to wind impact naturally yield higher intensities for cyclone types classified as windy. Therefore, I am not fully convinced that it is meaningful to use the same intensity measures both to define the cyclone groups and subsequently to assess the influence of different precursors on those same intensity measures. The differences between cyclone groups and their associated intensity measure requires further clarification.
I do appreciate the focus on distinguishing different cyclone types and exploring how their intensity is controlled by different precursors. It might be more straightforward to compare

the windy and rainy cyclone groups using an independent intensity measure, for example one based on vorticity, rather than relying on the same measures used for the grouping itself. Also, I am wondering whether the use of five different cyclone intensity measures is necessary. Would a smaller set, for example one wind based, one precipitation based, and one dynamical measure, be sufficient?

**Specific comments**
Title: 'controls of extratropical cyclone intensity': Maybe add something to also emphasize that different intensity types are considered as this is the focus of the study. For example, 'controls of extratropical cyclone intensity types/ groups.
Line 46ff: What is the point you want to make here? I am not sure the second half of the sentence is needed, or it needs to be stated how the coupled lower-level features are affected. This whole paragraph may be shortened.
Line 202: As you find that the genesis time has limited explanatory value on cyclone intensity, it would be great to justify why you choose to use genesis time instead of any other time. This could also be added in the introduction.
Line 206f: Would it be possible to provide a range between genesis time and maximum intensity?
Line 230: Since most of the cyclones are either in Dry+Calm or in Rainy+Windy, is there a distinction between impactful and non-impactful cyclones?
Line 320: Here suddenly the following discussion is about the cyclone groups and their climatology background. This discussion could be moved further down, after the sensitivity of the cyclone groups was discussed.
Line 379: With 'Precipitation' you mean PRECIP?
Line 418: One could also argue that the sensitivity differs for ETCs with different intensities, considering that in Section 3.1 sensitivities for different intensity measures are found. Here the wording intensity could mean both, please specify.
Line 451: It's great that MSLPa is included here, to provide another dynamical intensity measure!
Line 469f: 'The decrease in VO is almost as fast as the increase'. This sentence needs clarification on what increase is meant.
Line 605: Please specify here, what kind of 'ETC intensity' is meant, the intensity in terms of intensity measures or the intensity in terms of type of impact, i.e. windy/rainy?